# Three-Dimensional Airway Assessment as Diagnostic Aid in Obstructive Sleep Apnea

**DOI:** 10.3390/pathophysiology32040050

**Published:** 2025-09-26

**Authors:** Oscar Arturo Benítez-Cárdenas, Néstor Oliver Herrera-Salguero, Elhi Manuel Torres-Hernández, Miguel Angel Noyola-Frías, Ricardo Martínez-Rider, Marlen Vitales-Noyola

**Affiliations:** 1Service of Oral and Maxillofacial Surgery, Regional High Specialty Hospital “Dr. Ignacio Morones Prieto”, San Luis Potosi 78290, Mexico; oscar.benitez@uaslp.mx (O.A.B.-C.); manuel.torres@uaslp.mx (E.M.T.-H.); manf001@uaslp.mx (M.A.N.-F.); 2Department of Oral Surgery, Faculty of Dentistry, Autonomous University of San Luis Potosí, San Luis Potosí 78290, Mexico; nestor_herreracmf@hotmail.com (N.O.H.-S.); rmrider@uaslp.mx (R.M.-R.); 3Endodontics Postgraduate Program, Faculty of Dentistry, Autonomous University of San Luis Potosí, San Luis Potosí 78290, Mexico

**Keywords:** obstructive sleep apnea syndrome, cone-beam computed tomography, 3D airway evaluation, diagnostic imaging, airway volume

## Abstract

**Background**: Obstructive Sleep Apnea Syndrome (OSAS) is a prevalent and underdiagnosed condition with significant systemic and quality-of-life impacts. While polysomnography remains the gold standard for diagnosis, cone-beam computed tomography (CBCT) presents a potential adjunctive imaging tool for anatomical airway evaluation. **Objective**: We aimed to assess the effectiveness of three-dimensional airway evaluation via CBCT as a complementary diagnostic tool for OSAS. **Methods**: A diagnostic test study (experimental pilot study) was conducted using CBCT scans of 30 patients, divided into two groups: 15 scans from patients with a confirmed OSAS diagnosis through polysomnography and 15 scans from healthy controls. Five tomographic variables were analyzed: anteroposterior distance, lateral distance, minimum cross-sectional area, airway volume, and airway shape. Statistical analysis was performed comparing both groups. **Results**: The minimum cross-sectional area and airway volume showed statistically significant differences between the OSAS and control groups (*p* = 0.038 and *p* = 0.0055, respectively). Anteroposterior and lateral distances showed trends toward significance but were not statistically significant. **Conclusions**: CBCT-based airway analysis, particularly focusing on volumetric and cross-sectional area parameters, demonstrates strong potential as a complementary tool in the diagnosis of peripheral-type OSAS. However, it cannot replace polysomnography, especially for central OSAS diagnosis.

## 1. Introduction

Obstructive Sleep Apnea Syndrome (OSAS) is a sleep-related breathing disorder characterized by recurrent episodes of partial or complete obstruction of the upper airway during sleep, resulting in intermittent hypoxia, hypercapnia, sleep fragmentation, and excessive daytime sleepiness [1]. The pathophysiology of OSAS is multifactorial, involving both anatomical and functional components that contribute to airway collapse, particularly during rapid eye movement (REM) in sleep [2]. OSAS is associated with a wide range of significant comorbidities that contribute to its overall burden on public health. Cardiovascular complications such as systemic hypertension, coronary artery disease, arrhythmia, heart failure, and stroke which have been strongly linked to the intermittent hypoxia and sympathetic overactivation characteristic of OSAS. In addition, neurocognitive dysfunction, including impaired memory, decreased attention span, and executive dysfunction, is commonly reported, affecting both occupational and social functioning. Metabolic disturbances, particularly insulin resistance, type 2 diabetes mellitus, and dyslipidemia, are also frequently observed and may be exacerbated by the chronic systemic inflammation and hormonal dysregulation associated with disrupted sleep architecture. Collectively, these complications not only increase morbidity and mortality but also contribute to a markedly reduced quality of life in affected individuals. The clinical recognition of sleep-disordered breathing has evolved significantly over recent decades [3,4]. Early accounts identified the constellation of obesity, hypersomnolence, and respiratory disturbances during sleep, but it was not until the 1970s that Obstructive Sleep Apnea Syndrome (OSAS) was systematically defined as a distinct clinical entity. This shift was catalyzed by the identification of measurable physiological indicators, most notably the apnea–hypopnea index (AHI), which enabled objective diagnosis and facilitated epidemiological and interventional research on this increasingly prevalent condition [5]. As the understanding of OSAS advanced, so did the development of diagnostic methodologies capable of capturing its complex physiological manifestations. The gold standard for diagnosing Obstructive Sleep Apnea Syndrome (OSAS) remains nocturnal polysomnography (PSG), which provides comprehensive data on respiratory effort, airflow, oxygen saturation, electroencephalography (EEG) for sleep staging, and other physiological parameters essential for an accurate diagnosis [6]. PSG enables the calculation of the apnea–hypopnea index (AHI), which is the primary metric used to determine OSAS severity. However, despite its high diagnostic accuracy and comprehensive nature, PSG is expensive, time-consuming, requires specialized equipment and trained personnel, and often involves inconvenient overnight stays in sleep laboratories. These factors limit its availability, particularly in low-resource and rural settings, and contribute to significant underdiagnosis of OSAS worldwide. Consequently, there is growing interest in developing and validating alternative or complementary diagnostic modalities that are more accessible, cost-effective, and patient-friendly, including home sleep apnea testing (HSAT) and advanced imaging techniques such as cone-beam computed tomography (CBCT) for airway evaluation [6,7,8].

In this context, radiological imaging techniques have gained prominence as complementary tools in the assessment of patients with suspected Obstructive Sleep Apnea Syndrome (OSAS). Among these, cone-beam computed tomography (CBCT) has emerged as a particularly valuable modality due to its ability to provide detailed three-dimensional visualization of craniofacial anatomy and upper airway morphology. Recent studies highlight CBCT as a complementary tool for assessing airway morphology in OSAS, with consensus on its diagnostic value but ongoing debate about standardization and clinical applicability [9]. Initially designed for dental and maxillofacial applications, CBCT offers significant advantages over traditional computed tomography (CT), including lower radiation exposure, faster scan times, and cost-effectiveness, making it more accessible in clinical settings. In the evaluation of OSAS, CBCT facilitates precise identification and quantification of anatomical risk factors that contribute to airway obstruction, such as mandibular retrognathia, macroglossia, maxillary constriction, and pharyngeal airway narrowing. These anatomical insights are critical for individualized diagnosis, treatment planning, and surgical intervention considerations, particularly in patients where polysomnography findings and clinical symptoms are incongruent or when anatomical abnormalities are suspected to play a primary role [9,10,11]. Airway assessment through CBCT has demonstrated significant clinical utility in the detailed evaluation of upper airway morphology [10]. This imaging modality allows for precise quantification of key parameters, including total airway volume, minimum cross-sectional area, anteroposterior and lateral dimensions, as well as the overall shape and patency of the airway. Such measurements provide critical anatomical data that enhance our understanding of airway obstruction patterns specific to Obstructive Sleep Apnea Syndrome (OSAS). Multiple studies, including those by Ogawa et al., Abramson et al., and Schendel et al., have underscored the value of CBCT-derived airway metrics in distinguishing OSAS patients from healthy controls, demonstrating significant differences in airway size and configuration [12,13,14]. Moreover, the high-resolution three-dimensional visualization afforded by CBCT supports not only accurate diagnosis but also facilitates individualized treatment planning, particularly surgical interventions aimed at enlarging or stabilizing the airway. The ability to visualize and quantify airway structures in a non-invasive, reproducible manner makes CBCT a promising complementary tool alongside polysomnography and clinical evaluation in comprehensive OSAS management.

Given the rising prevalence of Obstructive Sleep Apnea Syndrome (OSAS) worldwide and the ongoing challenges associated with timely and accessible diagnosis, there is a pressing need to explore alternative or complementary diagnostic approaches.

This study seeks to evaluate the diagnostic utility of cone-beam computed tomography (CBCT)-based three-dimensional airway measurements in differentiating patients with OSAS from healthy individuals. By systematically correlating these tomographic airway parameters with the clinical diagnosis established by nocturnal polysomnography (PSG), we aim to determine the potential of CBCT as a reliable, non-invasive, and accessible adjunct diagnostic tool. Such an approach could be particularly valuable in resource-limited settings where conventional PSG is not readily available, thereby facilitating earlier detection, individualized treatment planning, and ultimately improved patient outcomes. The specific objectives of this study are to compare CBCT-derived airway volumetric and cross-sectional measurements between patients with OSAS and healthy controls; to analyze the diagnostic accuracy of CBCT airway parameters using ROC curve analysis against PSG as the reference standard; and to assess the feasibility of CBCT as a complementary diagnostic method in clinical settings with limited access to PSG. Based on these objectives, our research hypothesis is that CBCT-based three-dimensional airway measurements can reliably distinguish patients with OSAS from healthy individuals, providing a valid and accessible diagnostic adjunct to PSG. So, this pilot study addresses the research gap regarding the diagnostic utility of CBCT-derived airway measurements in distinguishing OSAS patients from healthy individuals.

## 2. Materials and Methods

### 2.1. Study Population and Design

This was a cross-sectional diagnostic accuracy study conducted with cone-beam computed tomography (CBCT) to evaluate structural differences in the upper airway between individuals with Obstructive Sleep Apnea Syndrome (OSAS) and healthy controls. A total of 30 CBCT scans of the head and neck region were retrospectively selected by non-probability sampling and divided into two equal groups: OSAS group (*n* = 15), patients with a prior confirmed diagnosis of OSAS based on overnight polysomnography. Control group (*n* = 15), individuals without clinical signs, symptoms, or prior diagnosis of sleep-disordered breathing. This was a pilot diagnostic study; therefore, no formal a priori power/sample-size calculation was performed. Participants were between 20 and 50 years of age, and all scans included met the criteria for appropriate anatomical coverage and image quality. Exclusion criteria comprised: history of maxillofacial or pharyngeal surgery, incomplete clinical records and CBCT scans of insufficient quality for airway analysis. Participants were recruited at the Faculty of Stomatology, Autonomous University of San Luis Potosi, Mexico. Recruitment was carried out consecutively from 2020 to 2021. All obtained data were included in Table 1. This study was approved by the Ethics Committee of the Faculty of Dentistry at the Autonomous University of San Luis Potosí (approval code: CEIFE-022–021 and approval date: 11 May 2021). All CBCT data were anonymized and processed in accordance with institutional and ethical research standards. Informed consent was waived due to the retrospective nature of the study and the use of previously acquired imaging data.

### 2.2. CBCT Imaging and Airway Analysis

All CBCT scans were obtained retrospectively from the imaging database of the Radiology Department at the Autonomous University of San Luis Potosi. All images had been acquired following standardized head and neck imaging protocols, ensuring consistency in patient positioning and image quality. Airway analysis was conducted using OnDemand 3D^®^ v.1.0.10.538 and NemoFAB^®^ software v22.0 platforms, which allow for three-dimensional segmentation and quantitative evaluation of upper airway structures, by using a semi-automatic thresholding method to delineate the upper airway boundaries, followed by manual editing to refine the contours and remove artifacts. The following anatomical variables were measured: Anteroposterior (AP) diameter: Linear distance between the posterior pharyngeal wall and the nearest point on the anterior wall at the narrowest level of the airway. Lateral diameter: Transverse width of the airway at its most constricted region. Minimum cross-sectional area: The smallest axial surface area of the airway, expressed in mm^2^. Airway volume: Total volume of the segmented upper airway region, measured in cubic centimeters (cm^3^). Airway shape classification: Qualitative assessment of airway morphology based on cross-sectional shape at the narrowest level (e.g., elliptical, round, square, irregular). All measurements were performed by a calibrated examiner, blinded to the clinical diagnosis, to minimize potential measurement bias. All scans were obtained using a NewTom VGi CBCT unit (NewTom VGi, Verona, Italy). The acquisition protocol was standardized with a voxel size of 0.3 mm, a field of view (FOV) of 15 × 15 cm, and exposure parameters of 110 kV, 0.58–1.49 mA, 1.76 mGy, and 3.6 s. Patients were positioned in natural head posture, with maximum intercuspation and without swallowing during image acquisition.

### 2.3. Statistical Analysis

Descriptive statistics were calculated for demographic and tomographic variables. Data normality was calculated by the Shapiro–Wilk test. Group comparisons tests were performed using t-tests or Mann–Whitney U tests, depending on normality assumptions. Categorical variables, including airway shape classification, were analyzed using the Chi-square test. For the statistical validation of CBCT measurements as an auxiliary diagnostic tool, the sensitivity and specificity of each tomographic variable were calculated using receiver operating characteristic (ROC) curve analysis. All tests were performed using GraphPad Prism v5.0 software (GraphPad Software, San Diego, CA, USA). Statistical significance difference was defined as *p* < 0.05. In addition to *p*-values, effect sizes were calculated to provide a more comprehensive interpretation of the results. For variables analyzed with Student’s t-test, effect sizes were reported as Cohen’s d. For non-parametric comparisons using the Mann–Whitney U test, effect size r was calculated using the formula Z/√n. Effect sizes were interpreted following conventional thresholds (small, medium, large) to complement statistical significance with the magnitude of the observed differences.

## 3. Results

### 3.1. Clinical and Demographic Data

All measurements from OSAS patients and controls were performed as described in material and methods section. Clinical and demographic data from patients and controls was registered in Table 1.

**Table 1 pathophysiology-32-00050-t001:** Clinical and demographic data from OSAS patients and controls.

Measurement	OSAS Patients	Controls	Effect Size Direction OSAS-Control
**Gender** (**%**) (Female/Male)	(40/60)	(53/47)	N/A
**Age** (**Years**)	36.5 ± 7.6	31.3 ± 7.2	N/A
**Systemic condition** (**%**)			N/A
Overweight/obesity	40	0	
Arterial hypertension	26.6	6.6	
Diabetes	6.6	6.6	
Hypertrigliceridemia	6.6	0	
**Tomographic measurements**			AUC ≈ 0.810, rank-biserial r/Cliff’s δ ≈ 0.619, medium–large effect Cohen d = 0.72, medium–large effect AUC ≈ 0.702, rank-biserial r/Cliff’s δ ≈ 0.404, medium–large effect AUC ≈ 0.807, rank-biserial r/Cliff’s δ ≈ 0.613, large effect
**Anteroposterior distance** (**mm**)	6.5, 5.5–13.3	10.9, 6.7–12.2	
**Lateral distance** (**mm**)	16.4 ± 6.6	21.9 ± 8.8	
**Minimum cross-sectional area** (**mm^2^**)	114.8, 76.09–146.9	161.1, 126.5–220.3	
**Volume** (**cm^3^**)	15.19, 12.5–20.3	22.3, 17.4–25.3	
**Shape of the pharynx** (**%**)			
Round	33.3	13.3	
Elliptical	13.3	26.6	
Square	13.3	33.3	
Anomalous	40	26.6	

Measurements are expressed in millimeters (mm), square millimeters (mm^2^), and cubic centimeters (cm^3^). N/A = not applicable. Data are represented as mean ± standard deviation or median and interquartile range.

### 3.2. Tomographic Measurements from OSAS Patients vs. Controls

All measurements were performed on patients and controls, as described in the Materials and Methods section. We do not observe significant difference in the values of the anteroposterior distance from patients and controls (*p* = 0.39, AUC ≈ 0.810, rank-biserial r/Cliff’s δ ≈ 0.619, medium–large effect) (Figure 1A), similar results were observed in lateral distance, however, there are a trend to increase this value in patients (*p* = 0.06, Cohen d = 0.72 medium–large effect) (Figure 1B). With respect to the minimum cross-sectional area, we found a significant difference in patients in comparison to controls (*p* = 0.038, AUC ≈ 0.702, rank-biserial r/Cliff’s δ ≈ 0.404, medium–large effect) (Figure 1C), where patients show increase values, in addition, in the parameter of volume, similar results were observed, (*p* = 0.0055, AUC ≈ 0.807, rank-biserial r/Cliff’s δ ≈ 0.613, large effect) (Figure 1D). Additional analyses were performed, classifying OSAS patients with and without systemic conditions, and no statistically significant differences were observed across any of the airway measurements evaluated, including anteroposterior distance, lateral distance, minimum cross-sectional area, and airway volume (*p* > 0.05) (Figure 2).

### 3.3. Tomographic Measurements as an Auxiliay Diagnosis Tool

ROC curves were performed for the assessment of the discriminatory capacity of each parameter to differentiate between healthy subjects and patients with OSAS. Regarding tomographic measurements, the anteroposterior distance was slightly lower in the OSAS group compared to the controls, although this difference was not statistically significant (*p* = 0.503, AUC = 0.58). The lateral distance showed a tendency toward smaller values in OSAS patients compared to controls, without reaching statistical significance (*p* = 0.061, AUC = 0.68). The minimum cross-sectional area was also reduced in the OSAS patients in comparison to controls, with no significant difference (*p* = 0.494; AUC = 0.72). Airway volume presented a similar pattern, being lower in OSAS patients compared with controls, showing a trend toward significance (*p* = 0.074; AUC = 0.80) (Figure 3). All data are shown in Table 2.

## 4. Discussion

The objective of this study was to evaluate whether cone-beam computed tomography (CBCT) can detect structural differences in the upper airway between individuals with Obstructive Sleep Apnea Syndrome (OSAS) and healthy controls. In this study, three-dimensional airway analysis using CBCT revealed statistically significant differences in airway volume and minimum cross-sectional area between patients with OSAS and control subjects, consistent with previous literature identifying these variables as relevant anatomical markers in the pathophysiology of the syndrome. Previous investigations, such as those by Ogawa et al. [12] and Abramson et al. [13], have also reported significant reductions in airway caliber in OSAS patients, particularly in volumetric dimensions and at the narrowest cross-sectional area. However, unlike some reports, in our series the minimum cross-sectional area values were higher in the OSAS group, which may be explained by anatomical variability, the patient’s position during image acquisition, or the relatively small sample size. Although anteroposterior and lateral distances did not show statistically significant differences, a trend toward increased lateral distance was observed in OSAS patients. This finding suggests that obstruction in this group may not depend exclusively on uniform narrowing, but rather on morphological changes that alter airway geometry without necessarily reducing linear diameters. This observation supports the value of three-dimensional evaluation over bidimensional measurements, as variations in shape and volume may go unnoticed in planar projections.

The clinical utility of these results lies in the fact that CBCT, while not replacing polysomnography (PSG) as the reference diagnostic method for OSAS, provides a valuable adjunct in the identification of structural risk factors for airway obstruction. By enabling precise three-dimensional visualization and quantification of airway dimensions, CBCT facilitates the detection of relevant anatomical alterations, such as pharyngeal narrowing, craniofacial skeletal discrepancies, or tongue base hypertrophy, which may predispose to or exacerbate obstructive events during sleep [15,16,17,18,19]. Early recognition of these structural features can inform timely referral for comprehensive sleep evaluation, including PSG, thereby reducing delays in diagnosis and treatment initiation. Furthermore, CBCT-derived airway metrics can be integrated into treatment planning for surgical interventions (e.g., maxillomandibular advancement, genioglossus advancement, or uvulopalatopharyngoplasty) and for the customization of mandibular advancement devices used in dental sleep medicine [20,21,22]. The capacity to simulate postoperative changes in airway volume and morphology enhances preoperative planning accuracy and may improve treatment outcomes. From a practical perspective, CBCT is widely available in dental and maxillofacial surgery settings and offers significant advantages over conventional multi-slice CT, including lower radiation dose, shorter acquisition time, and reduced cost [23,24]. These features make it particularly attractive in resource-limited environments where access to sleep laboratories is scarce and where a quick, accessible imaging modality can support early risk stratification. In such contexts, CBCT may serve as a “gatekeeper” tool, identifying patients who require further specialized testing while avoiding unnecessary referrals for individuals with low anatomical risk. Nevertheless, it is essential to emphasize that CBCT assesses static anatomical structures and does not capture dynamic functional changes during sleep, nor does it differentiate central from obstructive events. Therefore, its optimal use lies in complementing, rather than replacing, established diagnostic pathways. Future efforts to standardize CBCT airway measurement protocols and to correlate these with clinical and polysomnographic outcomes could enhance its role in integrated OSAS diagnostic strategies.

In addition, analysis classifying patients with and without systemic conditions did not find significant differences across any of the airway measurements evaluated, including anteroposterior distance, lateral distance, minimum cross-sectional area, and airway volume. This suggests that, within our sample, the presence of systemic comorbidities, such as overweight/obesity, hypertension, or metabolic disorders, did not produce measurable differences in the static anatomical airway parameters assessed by CBCT. Previous studies have reported variable associations between systemic conditions and airway dimensions in OSAS. For instance, obesity has been linked to increased soft tissue deposition in the upper airway, potentially reducing lumen size [25,26], while hypertension and metabolic syndrome have been more strongly associated with functional impairment, such as altered ventilatory control and endothelial dysfunction, rather than with overt structural narrowing detectable on static imaging [27,28]. Our findings may indicate that, in this population, the structural airway alterations associated with OSAS are predominantly determined by craniofacial skeletal configuration and pharyngeal morphology, rather than being substantially modified by systemic comorbidities. However, the relatively small sample size and the static nature of CBCT limit the ability to detect subtle differences that could emerge during dynamic airway collapse in sleep. Further research integrating three-dimensional anatomical assessment with functional studies, such as drug-induced sleep endoscopy or dynamic MRI, could clarify the extent to which systemic conditions modulate airway collapsibility in OSAS [29,30].

The ROC curves drawn for each anatomical parameter reveal varying levels of discrimination; the anteroposterior distance obtained an AUC of 0.58, a value barely above 0.5, which characterizes a test with no discriminatory capacity. Thus, this measure alone does not reliably distinguish OSAS patients from controls. The lateral distance presented an AUC of 0.68; according to the ROC curve classification, values between 0.7 and 0.8 are acceptable, so this result points to only moderate capacity. The cross-sectional area reached an AUC of 0.72, already within the acceptable range, although the differences did not reach statistical significance. The best performance corresponded to pharyngeal volume, with an AUC of 0.80, which falls into the range of excellent tests; this means that, in approximately 80% of comparisons, an OSAS patient will have a smaller volume than a healthy subject. This finding is consistent with the pathophysiology of OSAS, where a narrow upper airway and reduced volume favor collapsibility during sleep [31]. Even so, none of the variables reached conventional statistical significance, suggesting that these parameters could be useful as complementary markers but do not replace standard diagnostic methods. Studies with larger sample sizes are required to confirm the usefulness of these measures.

The limitations of this study include the relatively small sample size, its retrospective design, and the absence of correlation with functional sleep parameters beyond prior polysomnography diagnosis. Additionally, the influence of head position or respiratory phase on measurements was not assessed, as these factors could affect the results. Another limitation is the use of a non-probability, consecutive sampling method, which may restrict the generalizability of the findings. Despite these barriers, this study provides preliminary evidence supporting the feasibility of CBCT as a complementary diagnostic tool for OSAS. Future studies with larger and more diverse samples, prospective designs, standardized acquisition protocols, and integrated analysis of both anatomical and functional variables will help refine the role of CBCT in the diagnostic approach to OSAS. Moreover, multicenter collaborations and the inclusion of longitudinal follow-up could strengthen external validity and clinical applicability. As a pilot study, the present work highlights the potential clinical impact of CBCT-derived airway measurements in improving early detection and individualized treatment planning, particularly in resource-limited settings where access to full polysomnography may be restricted. Unresolved issues remain in current research, including the unclear correlation between CBCT parameters and OSAS severity (AHI classification), the limited exploration of CBCT differences across OSAS subtypes, and the insufficient validation of its predictive value for treatment outcomes such as surgery or oral appliances.

Overall, our findings support the incorporation of three-dimensional airway evaluation into imaging protocols for patients with suspected or confirmed OSAS, promoting interdisciplinary collaboration among dentists, maxillofacial surgeons, otolaryngologists, and sleep medicine specialists.

## 5. Conclusions

CBCT-based three-dimensional (3D) airway evaluation demonstrates substantial diagnostic value in the context of obstructive sleep apnea syndrome (OSAS), particularly through its ability to quantify volumetric and cross-sectional airway dimensions with high anatomical accuracy. While CBCT is not intended to replace polysomnography, the gold standard for OSAS diagnosis, it serves as a valuable complementary tool that can aid in the early detection of anatomical airway restrictions associated with the condition. The integration of 3D airway analysis into routine maxillofacial imaging protocols may facilitate interdisciplinary collaboration among dental specialists, sleep physicians, and otolaryngologists, improving diagnostic workflows and informing personalized treatment planning. Clinically, this approach has the potential to enhance screening in patients at risk for OSAS, especially those undergoing imaging for other craniofacial concerns, thereby contributing to earlier intervention, better patient outcomes, and reduced burden of undiagnosed sleep apnea. Further research and standardization of CBCT airway assessment protocols are encouraged to refine its clinical applications and reinforce its role in comprehensive OSAS evaluation strategies. In conclusion, CBCT-derived airway measurements showed significant differences between OSAS patients and controls, demonstrated diagnostic accuracy through ROC analysis, and proved feasible as a complementary tool to PSG in clinical settings.

## Figures and Tables

**Figure 1 pathophysiology-32-00050-f001:**
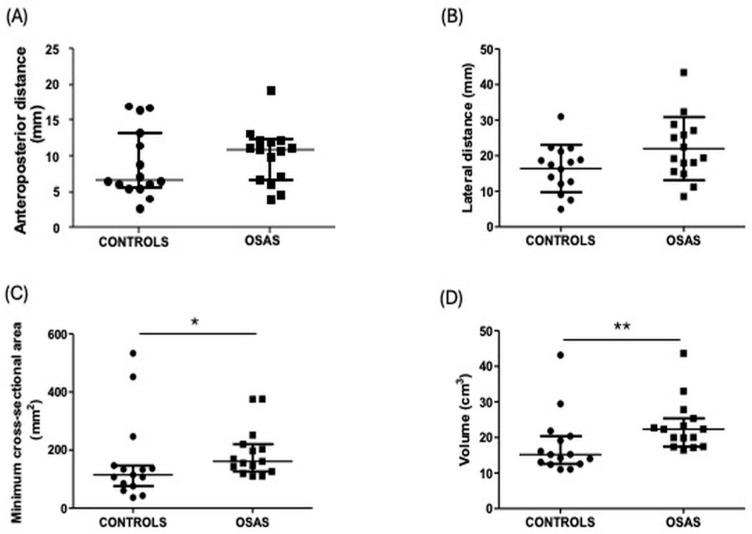
Tomographic measurements from controls vs. OSAS patients. (**A**) Anteroposterior distance in mm from controls and patients. Data are represented as median and interquartile range. (**B**) Lateral distance in mm from controls and patients. Data are represented as mean and standard deviation. (**C**) Minimum cross-sectional area in mm^2^ from controls and patients. Data are represented as median and interquartile range. (**D**) Volume in cm^3^ from controls and patients. Data are represented as median and interquartile range. *n* = 15. Mm = millimeters, mm^2^ = square millimeters, cm^3^ = cubic centimeters. * *p* < 0.05; ** *p* < 0.01.

**Figure 2 pathophysiology-32-00050-f002:**
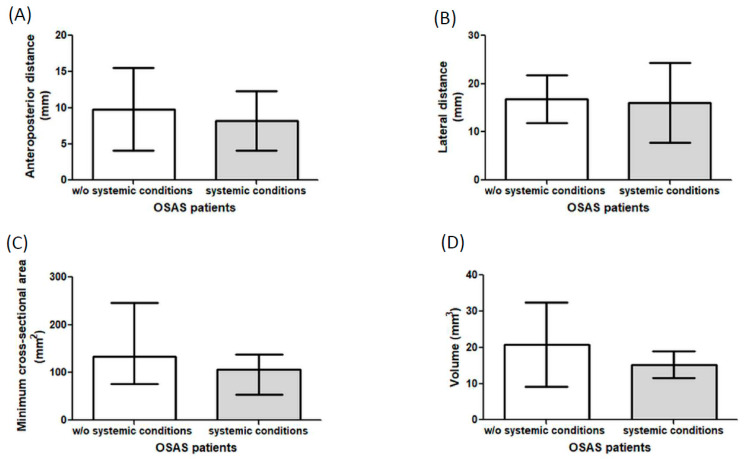
Tomographic measurements from OSAS patients classified as w/o systemic conditions and with systemic conditions. (**A**) Anteroposterior distance in mm from OSAS patients without and with systemic conditions. (**B**) Lateral distance in mm from OSAS patients without and with systemic conditions. (**C**) Minimum cross-sectional area in mm^2^ from OSAS patients without and with systemic conditions. (**D**) Volume in cm^3^ from OSAS patients without and with systemic conditions. Data are represented as mean and standard deviation or median and interquartile range. *n* = 15. w/o = without. Mm = millimeters, mm^2^ = square millimeters, cm^3^ = cubic centimeters.

**Figure 3 pathophysiology-32-00050-f003:**
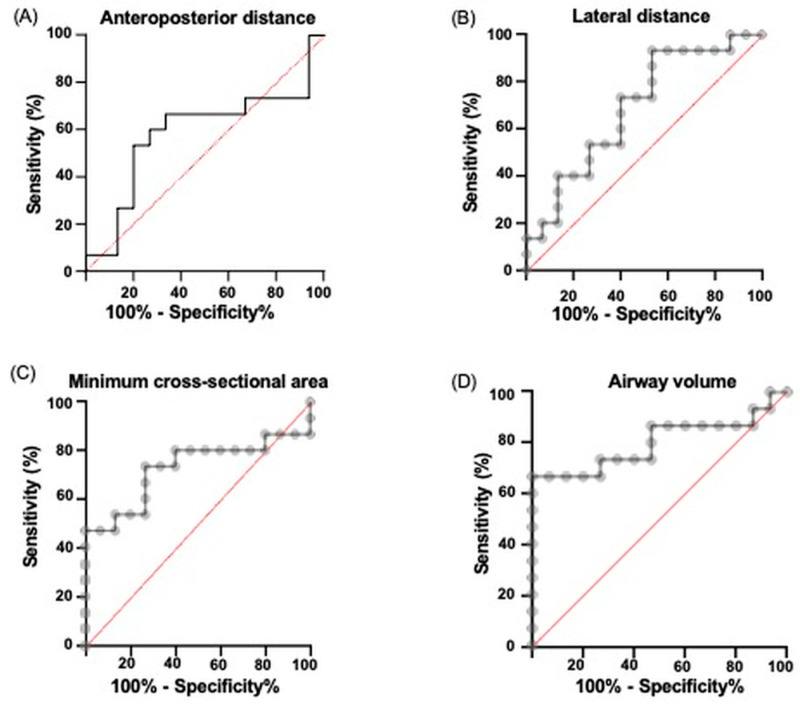
Receiver Operating Characteristic (ROC) curves for tomographic variables in patients with OSA and controls. ROC curves show the diagnostic performance of (**A**) anteroposterior distance, (**B**) lateral distance, (**C**) minimum cross-sectional area, and (**D**) pharyngeal volume. The pharyngeal volume demonstrated the highest discriminative ability (AUC = 0.80), followed by minimum cross-sectional area (AUC = 0.72), lateral distance (AUC = 0.68), and anteroposterior distance (AUC = 0.58).

**Table 2 pathophysiology-32-00050-t002:** Area under the curve (AUC) of tomographic variables in OSAS patients and controls.

Measurement	*p* Values	Area Under Curve (AUC)
Anteroposterior distance	0.503	0.58
Lateral distance	0.061	0.68
Minimum cross-sectional area	0.494	0.72
Airway volume	0.074	0.80

## Data Availability

All data supporting the findings of this study are contained within the article. No additional data are available.

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
