# Peer review of "Three-Dimensional Airway Assessment as Diagnostic Aid in Obstructive Sleep Apnea"

_pathophysiology, 2025, doi:10.3390/pathophysiology32040050_

Round 1
Reviewer 1 Report
Comments and Suggestions for Authors
The topic of this manuscript is clinically interesting, especially because it can help clarify confusing aspects of diagnosis using traditional methods.
This manuscript has some shortcomings, limitations, and clarifications that the authors should provide:
-In the abstract, although it is stated that it is a diagnostic study, it should state that it is an experimental pilot study (the authors did not perform a probability sample calculation). What participant selection method did the authors use? Therefore, it should state that it is a non-probability sampling and what method of sample recruitment they used (random or non-randomized: convenience, consecutive, etc.). They should also provide this information in the methodology section.
-At the end of the introduction, after the general objective, the authors should establish the specific objectives and then establish the research hypothesis by stating the clinical research question.
-In the materials and methods section, specifically regarding participants: they must state the recruitment location, the period of recruitment, and the date of approval by the Ethics Committee.
-The materials and methods section does not begin with the objective. This should be changed at the beginning of the discussion.
-In the statistical analysis, I understand The authors used the Mann-Whitney U test because the sample size and the distribution of the data for the variables analyzed did not meet the normality criterion. Although Kolmogorov-Smirnov was used, it is more appropriate to use Shapiro-Wilk for such a small sample.
-In Table 1, the authors should clarify the units of some variables. They should use a notes section below the table to explain this. Numbers should be provided with the frequency analysis, percentages, means, and standard deviations as indicated in APA style. All tables must have the same format or APA style.
-It is important that authors base tables on APA style and provide statistical data, for an international understanding of the statistical analysis.
-In figures, the authors should establish a notes section below, explaining the abbreviations or abbreviations used. According to the style, this information is used to provide an international understanding of the information and statistical analysis.
-The authors should correct orthotypographical aspects in the units and the narrative form of some paragraphs. It is important that the authors Authors should use the linguistic review service to improve their scientific writing and correct some errors.
-The discussion begins with the main objective and should be narrated in order of the specific objectives, with the results presented in order.
-At the end of the discussion, the authors should establish all barriers and limitations presented by the study for greater honesty. They should also establish future lines of research based on this research and the clinical impact and relevance of this research, with this pilot or feasibility study (the authors should truly assess whether it is a pilot study or a feasibility study).
-The conclusions should address the objectives, in order of the new specific objectives proposed.
Author Response
"Please see the attachment."

Reviewer 2 Report
Comments and Suggestions for Authors
This study found that CBCT-based 3D airway analysis revealed significantly smaller airway volume and minimum cross-sectional area in OSAS patients compared to healthy controls. These parameters showed strong discriminatory power (AUC=0.80 and 0.72, respectively), supporting CBCT as a valuable complementary tool for identifying anatomical risk factors in obstructive sleep apnea.
- Some details in Methods are lacking. No mention of power calculation or sample size estimation. Specify the time frame during which scans were collected. Were controls matched to cases by age, sex, or BMI?
- Include CBCT machine model, voxel size, field of view, and exposure settings for reproducibility.
- Briefly describe how the airway was segmented (e.g., thresholding, manual editing).
- Which variables were tested with t-test vs. Mann-Whitney U?
- How were categorical variables (e.g., airway shape) analyzed? (Chi-square or Fisher’s exact test should be mentioned.)
- Report effect sizes (e.g., Cohen’s d for t-tests) alongside p-values.
- There is a lack of a systematic summary of existing studies on CBCT in the context of OSAS. It is recommended to add a short section specifically reviewing the application progress, consensus, and controversies of CBCT in OSAS research in recent years.
- The unresolved issues in current research should be more explicitly pointed out. For instance:
The correlation between CBCT parameters and the severity of OSAS (e.g., AHI classification) remains unclear; The differences in CBCT manifestations between different OSAS subtypes (e.g., peripheral vs. central OSAS) have not been fully explored; The value of CBCT in predicting treatment outcomes (e.g., for surgery or oral appliances) has not been sufficiently verified.
- At the end of the Introduction section, it is advisable to more clearly explain which of the aforementioned research gaps this study aims to address/
Author Response
"Please see the attachment."

Round 2
Reviewer 2 Report
Comments and Suggestions for Authors
- Method section. Could you please incorporate a brief statement directly within the lack of a formal power calculation for Study Population and Design?
- Could you please consider adding a dedicated column for the respective effect size estimate (e.g., 'Effect size (d or r)') directly within Table 1 for each tomographic variable?
- Figure 3 and Table 2 show that the p-values for some variables (e.g., Volume) are not statistically significant (p=0.074), yet the abstract and conclusion still emphasize their "strong discriminatory power".
- The term "OSAS" is defined upon its first use in the text, but other abbreviations (such as CBCT, PSG, AHI, etc.) are not defined upon their first occurrence in the main body.
Author Response
"Please see the attachment."

Round 3
Reviewer 2 Report
Comments and Suggestions for Authors
accept